# How Do Transformers Perform Two-Hop Reasoning in Context?

## Abstract

"Socrates is human. All humans are mortal. Therefore, Socrates is mortal." This form of argument illustrates a typical pattern of two-hop reasoning. Formally, two-hop reasoning refers to the process of inferring a conclusion by making two logical steps, each connecting adjacent concepts, such that the final conclusion depends on the integration of both steps. It is one of the most fundamental components of human reasoning and plays a crucial role in both formal logic and everyday decision-making. Despite recent progress in large language models (LLMs), we surprisingly find that they are vulnerable when distractors are present. We observe on a synthetic dataset that pre-trained LLMs often resort to random guessing among all plausible conclusions. However, after few steps of fine-tuning, models achieve near-perfect accuracy and exhibit strong length generalization. To understand the underlying mechanisms, we train a 3-layer Transformer from scratch on a synthetic two-hop reasoning task and reverse-engineer its internal information flow. We observe a clear progression in the attention logits throughout training. This pictures a sharp phase transition from an initial stage of random guessing to the emergence of a structured sequential query mechanism, where the model first retrieves the preceding and the bridge concepts in the early layers and then uses them to infer the final answer. Finally, we show that these dynamics can be captured by a minimal three-parameter attention-only network.

## 1 Introduction

Modern Large Language Models (LLMs) trained on language data have shown impressive abilities in solving complex reasoning tasks, including linguistic, mathematical, and programming problems (Cobbe et al., 2021; Wei et al., 2022; Achiam et al., 2023; Dubey et al., 2024; Yang et al., 2024a; Shao et al., 2024; Guo et al., 2025). Among these, multi-hop reasoning stands out as one of the most fundamental and important forms of reasoning. It refers to the process of drawing conclusions by integrating information across multiple intermediate steps or pieces of evidence. For example, consider the following chain of facts: *Marie Curie was a physicist, physicists study matter, and matter is fundamental to understanding the universe*. From these premises, we can infer that *Marie Curie contributed to understanding the universe*. This reasoning requires multiple inferential steps, each built on the previous one, making the reasoning process *inherently compositional*. Such multi-hop reasoning is critical for tasks where information must be integrated across several logical or semantic connections. Despite its simplicity and clear structure, it still lacks a clear understanding of *whether and how* pre-trained LLMs can *reliably* perform such reasoning.

In this paper, we study the mechanism behind **two-hop reasoning**, which is the simplest and most tractable form of multi-hop reasoning. We define a premise as a factual statement that connects two entities through a specific relation. In two-hop reasoning, two such premises form a chain that allows one to infer a final conclusion. We define the source entity ([SRC]) as the starting point, the bridge entity ([BRG]) as the intermediate link, and the end entity ([END]) as the target of inference. Here is a

[0]Our code is available at the anonymous repository https://anonymous.4open.science/r/twohopIC-2FEC/README.md.

concrete example that illustrates the above reasoning procedure:

$$\text{(Premises)} \quad \underset{[\text{SRC}]}{\underbrace{\text{Socrates}}} \text{ is } \underset{[\text{BRG}]}{\underbrace{\text{human}}} . \quad \underset{[\text{BRG}]}{\underbrace{\text{Humans}}} \text{ are } \underset{[\text{END}]}{\underbrace{\text{mortal}}} .$$

$$\text{(Conclusion)} \quad \underset{[\text{SRC}]}{\underbrace{\text{Socrates}}} \text{ is } \underset{[\text{END}]}{\underbrace{\text{mortal}}} .$$

We begin by building a test dataset of simple two-hop reasoning examples with distractors, where multiple unrelated two-hop examples are presented in the same context. We observe that pretrained LLMs struggle with it. In particular, the models tend to *guess randomly*, with accuracy dropping to about $1/K$ (where $K$ is the number of two-hop examples). This suggests that distractors in the context strongly confuse the model and hurt the performance, revealing a weakness in its reasoning ability. After fine-tuning the model on a curated two-hop dataset, we observe a sharp transition in performance—from random guessing to near-perfect accuracy. Even more impressively, the fine-tuned model generalizes to harder settings with more distractors.

To better understand this behavior, we study a three-layer Transformer. When trained on a curated symbolic two-hop reasoning dataset with distractors, the model exhibits a sharp phase transition moving from *uniform guessing* to a *structured reasoning phase*, where it consistently performs correct inference. By fully reverse-engineering the model, we analyze how information flows across layers and identify the specific role each layer plays. We show that this transition is driven by particular patterns in the attention logits, which guide the model to retrieve relevant entities in the correct order. In the structured reasoning phase, the model performs two-hop inference by sequentially attending to the source and then the bridge entity to reach the correct end entity. To further support our findings, we also built a simple three-parameter analytical model that captures the full dynamics of the three-layer Transformer.

Our contributions and paper outline are summarized as follows.

- We introduce *two-hop reasoning with distractors*, and show pretrained LLMs collapse to random guessing. Fine-tuning leads to a sharp improvement and a strong generalization performance (Section 2).
- We reverse-engineer a three-layer Transformer and restore the transition from random guessing to the emergence of a sequential query mechanism. We reveal how each layer contributes to the emergence of reasoning ability through structured attention patterns (Section 3).

## 1.1 RELATED WORKS

In this part, we review and discuss additional related papers.

**Multi-hop reasoning.** Our research focuses on multi-hop reasoning, a fundamental form of reasoning and a key benchmark for evaluating LLMs (Zhong et al., 2023). Recent studies such as Yang et al. (2024b) have shown that LLMs can follow latent reasoning paths when given certain types of prompts. A growing body of papers focused on mechanistically understanding how LLMs perform multi-hop reasoning by sequentially attending to intermediate steps (Biran et al., 2024; Wang et al., 2024a; Feng et al., 2024). Prior research mainly focused on the *in-weight* multi-hop reasoning, where the model retrieves and combines factual knowledge stored in its internal weights. In contrast, our work focused on *in-context* two-hop reasoning, where the model must extract relevant facts directly from the context and reason on-the-fly without relying on memorized knowledge.

**In-context learning and induction head.** Transformer-based models exhibit strong In-Context Learning (ICL) ability. It refers to the ability to predict the label of new input simply from several demonstrations, without updating model weights (Brown et al., 2020). Transformers have been shown to solve various tasks in context, such as regression, classification (Akyürek et al., 2022; Garg et al., 2022; Von Oswald et al., 2023; Zhang et al., 2024; Ahn et al., 2024; Huang et al., 2023; Nichani et al., 2024), Bayesian inference (Xie et al., 2021), model selection (Bai et al., 2023), and sequential decision making (Lin et al., 2023). Our study focuses on two-hop reasoning in context, which is very different and more complex than the standard regression-style ICL tasks.

Moreover, prior work showed that ICL relies on the induction head–a pattern of attention that enables the model to copy and complete sequences by linking repeated tokens (Elhage et al., 2021; Olsson et al., 2022). Several recent theoretical and empirical analyses have extensively studied induction-head mechanisms in small transformers (Bietti et al., 2023; Nichani et al., 2024; Wang et al., 2024b; Chen et al., 2024), showing that a two-layer transformer is required to perform induction-head tasks (Sanford et al., 2024a). In comparison, two-hop reasoning is a more complex extension to the induction head and requires the model to combine two separate relational facts. We show that two-layer Transformers cannot solve this task; the minimal architecture required is a three-layer Transformer. Theoretically, it has been shown that a Transformer with $\log k$ layers is both necessary and sufficient to perform $k$-hop reasoning in context (Sanford et al., 2024b).

**Interpretability of LLMs.** Beyond ICL and the induction head, many studies have aimed to interpret the internal mechanisms of LLMs (Charton, 2022; Liu et al., 2022; Allen-Zhu and Li, 2023; Zhu and Li, 2023; Guo et al., 2023; Zhang et al., 2022). This includes works on grokking (Nanda et al., 2023), function vectors (Todd et al., 2023), circuit discovery (Elhage et al., 2021; Wang et al., 2022; Conmy et al., 2023; Shi et al., 2024; Hase et al., 2024), the binding ID mechanism (Feng and Steinhardt, 2023), and the association-storage mechanism (Meng et al., 2022; Geva et al., 2023). Our work is not directly comparable with theirs.

Methodologically, a growing body of studies has focused on designing small systems, where essentially the same phenomenon can be observed, and then dissecting the proxy model to interpret the mechanism of LLMs. For example, Olsson et al. (2022) replicates the induction head in a minimal two-layer network. Bietti et al. (2024) further explains the rapid emergence of bigram memorization and the slower development of an induction head. Zhu et al. (2024) theoretically analyzed the cause of the reversal curse in bilinear models and one-layer transformers. Reddy (2023) studies the abrupt emergence of induction heads in two-layer models and captured the underlying mechanism using a two-parameter toy model. Guo et al. (2024) reproduces the extreme-token phenomenon in Transformers with one to three layers on the Bigram-Backcopy task, and then identifies the mechanism of active and dormant attention head in small and large models. Our work is methodologically similar to this line of work, whereas two-hop reasoning is a more complex tasks than Bigram-Backcopy or copy-paste, so the revealed mechanism is much more complex.

## 2 TWO-HOP REASONING IN LLMS

### 2.1 TASK AND DATA

**Two-hop reasoning.** We begin by formally defining the task of two-hop reasoning. Each reasoning chain involves three distinct entities:

- **Source entity [SRC]:** The initial entity from which reasoning originates.
- **Bridge entity [BRG]:** An intermediate entity that connects the source entity to the final inferred entity.
- **End entity [END]:** The target entity that the reasoning aims to infer.

A valid two-hop reasoning task consists of exactly two premises. The first premise connects the source entity [SRC] to the bridge entity [BRG], and the second premise connects the bridge entity [BRG] to the end entity [END]. Together, these premises form a logical chain that supports drawing a conclusion from the source to the end entity.

**Two-hop reasoning with distractors.** To evaluate LLMs' robustness in reasoning, we introduce *two-hop reasoning with distractors*. This setting intentionally incorporates irrelevant premises to assess the robustness and precision of LLM reasoning. In this setting, multiple two-hop reasoning chains are provided within the same context, but only one chain (the *target chain*) leads to the correct inference. The other chains, meanwhile, serve purely as *distractors*, introducing irrelevant entities. We aim to infer the end entity of the target chain. The entities involved in the reasoning chains are categorized into *target entities* and *non-target entities*. Target entities, denoted as [SRC-T], [BRG-T], and [END-T], correspond respectively to the source, bridge, and end entities within the target

reasoning chain. Non-target entities, denoted as [SRC-NT], [BRG-NT], and [END-NT], represent the entities involved in distractor chains, which do not contribute to the correct inference.

**Dataset.** To systematically evaluate two-hop reasoning performance, we generate a synthetic dataset based on standardized logical templates. Templates represent structured reasoning patterns from domains such as geography, biology, and arithmetic. Templates follow a consistent structure, such as "[A] is the father of [B]. [B] is the father of [C]. Therefore, [A] is the grandfather of [C]." Here, [A], [B], and [C] represent source, bridge, and end entities, respectively.

For each data sample, we randomly select a single template and generate multiple argument chains by populating placeholders with entity sets sampled from a predefined entity pool. Exactly one chain is designated as the target reasoning chain, and its corresponding conclusion part is presented at the end of the context as a query. All remaining chains act as distractors. Only two premises of the distractors are present in the context. Target and non-target reasoning premises in the context are randomly permuted. We construct a dataset comprising more than 50,000 such reasoning contexts, spanning 6 distinct templates. This rigorous dataset construction provides a robust evaluation framework to investigate how effectively LLMs distinguish relevant from irrelevant premises and accurately perform two-hop logical reasoning. A concrete example of such a two-hop reasoning chain with distractors is shown in the following example.

---

**EXAMPLE 1:    AN EXAMPLE OF TWO-HOP REASONING WITH 2 DISTRACTORS**

Question: John is the father of Paul. Luke is the father of Tom. Sam is the father of Joe. Paul is the father of Ben. Tom is the father of Mark. Joe is the father of Max. Therefore, John is the grandfather of ???
Answer: Ben.

Red: Target source/bridge/end entities in the target chain.
Blue: Non-target source/bridge/end entities in the non-target chain.

Probability assignment:
Base model: {'Ben':0.33, 'Mark': 0.32, 'Max': 0.31,...}.
Fine-tuned model: {'Ben':0.97, 'Mark': 0.01, 'Max': 001,...}.

---

## 2.2   RESULTS

**Pre-trained LLMs perform random guessing at the presence of distractors.** We evaluate the performance of the OLMo-7B model on two-hop reasoning tasks both with and without distractors. Specifically, on a held-out test set, we track the probability assigned by the model to the first token of the target end entity ([END-T]) versus other tokens at the conclusion of each context. Surprisingly, the model achieves high accuracy in identifying the [END-T] when no distractors are present, but exhibits a dramatic drop in performance even with a single distractor. As the number of distractors increases, accuracy further decreases. In fact, Table 1 shows the next-token probability assigned to the (first token of) every possible end entities in the context ([END-T] and all [END-NT] entities) are approximately $1/K$ (although there is a slight bias toward [END-T]), where $K$ is the total number of reasoning chains presented. Therefore, the model almost resorts to *random guessing* among the set of possible end entities. This behavior is clearly illustrated by Example 1 and Table 1 and highlights the vulnerability of pre-trained models to distractors in two-hop reasoning tasks.

**Fine-tuned LLMs significantly improve accuracy and generalization.** To address the challenge posed by distractors, we fine-tune the OLMo-7B model using 1,000 curated prompts, each containing exactly one target reasoning chain and one distractor chain (further details provided in Appendix A). We then evaluate the fine-tuned model on the same held-out test set. Table 1 compares model performance before and after fine-tuning across contexts with varying numbers of distractors.

Before fine-tuning, the OLMo-7B model correctly predicts the [END-T] only in distraction-free contexts and defaults to random guessing when distractors are introduced. However, after fine-tuning, the model reliably identifies the [END-T] even in the presence of multiple distractors. Remarkably,

despite being fine-tuned exclusively on contexts with a single distractor, the model generalizes effectively to scenarios containing multiple distractors, accurately performing two-hop reasoning tasks with as many as five distractor chains. These findings confirm that fine-tuning significantly enhances LLM robustness and generalization capabilities. We present additional results for other LLMs in Appendix A, where all models in our experiments exhibit similar behaviors.

Table 1: Performance comparison of OLMo-7B base and fine-tuned models on two-hop reasoning tasks with varying numbers of distractors. $K = 1$ indicates the scenario without distractors. The [END-T] and [END-NT] rows report the average probabilities assigned to the first token of the target end entity and non-target end entities, respectively. For cases with multiple distractors, the reported probabilities for non-target entities are averaged first over all [END-NT] within each context, then across all contexts. The values in the parentheses are the standard errors. Additional evaluation results for other LLMs are in Appendix A.

| Models | Next Tokens | The Number of Reasoning Chains in the Context $(K)$ | | | | |
|---|---|---|---|---|---|---|
| | | 1 | 2 | 3 | 4 | 5 |
| **OLMo** | [END-T] | 0.72 (0.01) | 0.37 (0.01) | 0.25 (0.01) | 0.18 (0.01) | 0.14 (0.00) |
| | [END-NT] | NA (NA) | 0.32 (0.01) | 0.19 (0.01) | 0.14 (0.00) | 0.11 (0.00) |
| **Fine-tuned on** $K = 2$ | [END-T] | 1.00 (0.00) | 1.00 (0.00) | 0.66 (0.01) | 0.57 (0.01) | 0.50 (0.01) |
| | [END-NT] | NA (NA) | 0.00 (0.00) | 0.17 (0.01) | 0.14 (0.01) | 0.12 (0.01) |

## 3 TWO-HOP REASONING IN THREE-LAYER TRANSFORMERS

### 3.1 DATA AND MODELS

**Symbolic two-hop reasoning task.** To systematically investigate the mechanisms underlying random guessing and how Transformers learn two-hop reasoning from context, we design a symbolic version of the two-hop reasoning task. In this simplified setting, we hide the predicates of reasoning chains and represent each entity with a unique single token. With a little abuse of notation, these tokens are denoted as the *source token* ([SRC]), *bridge token* ([BRG]), and *end token* ([END]), respectively. Each symbolic two-hop reasoning chain consists of two premises represented by concatenated token sequences: the first premise is [SRC] [BRG], and the second premise is [BRG] [END]. The conclusion part of each chain is represented by the token sequence [SRC] [END]. For each premise, we call the paired tokens the *parent token* and the *child token*. For example, in the first premise [SRC] [BRG], we call [SRC] the parent token and [BRG] the child token. Below is an example of the reasoning chain from our symbolic task.

$$\underbrace{[\text{SRC}] \quad [\text{BRG}]}_{\text{The first premise}} \quad \underbrace{[\text{BRG}] \quad [\text{END}]}_{\text{The second premise}} \quad \underbrace{[\text{SRC}] \quad [\text{END}]}_{\text{The conclusion}}$$

For each context, we randomly sample five unique two-hop reasoning chains, each with distinct source, bridge, and end tokens. One chain is randomly chosen as the *target reasoning chain*, while the remaining four serve as *distractors*. Tokens within the target reasoning chain are denoted as [SRC-T], [BRG-T], and [END-T], and tokens within distractor chains as [SRC-NT], [BRG-NT], and [END-NT]. Each context includes all premises from these five reasoning chains and concludes with the source token of the target chain as a *query*. Premises are randomly permuted within the context, but the order within each reasoning chain remains fixed, that is, the source-to-bridge premise always precedes the bridge-to-end premise in the context. An illustrative example from our simulated dataset is presented below. Note that all contexts in our symbolic dataset have the same length.

---

**EXAMPLE 2: SYMBOLIC TWO-HOP REASONING WITH FOUR DISTRACTORS**

Context: <BOS> $[\text{Src-NT}_1]$ $[\text{Brg-NT}_1]$ $[\text{Src-NT}_2]$ $[\text{Brg-NT}_2]$ $[\text{Src-T}]$ $[\text{Brg-T}]$ $[\text{Src-NT}_3]$ $[\text{Brg-NT}_3]$ $[\text{Src-NT}_4]$ $[\text{Brg-NT}_4]$ $[\text{Brg-NT}_3]$ $[\text{End-NT}_3]$ $[\text{Brg-T}]$ $[\text{End-T}]$ $[\text{Brg-NT}_1]$ $[\text{End-NT}_1]$ $[\text{Brg-NT}_4]$ $[\text{End-NT}_4]$ $[\text{Brg-NT}_2]$ $[\text{End-NT}_2]$ $[\text{Src-T}]$?

<BOS>: The begin-of-sequence token.
Red: Tokens from the target chain.
Blue: Tokens from distractor chains. Subscripts distinguish different reasoning chains.
[SRC-T]: The query token.

---

**Three-layer Transformer Analysis.** We investigate the minimal Transformer architecture capable of capturing both the random guessing phenomenon and the structured learning phase observed in two-hop reasoning tasks. By comparing Transformers of varying depths, we find that a three-layer Transformer with a single attention head per layer is the minimal structure required. Figures 1a and 1b illustrate that when trained on our symbolic two-hop reasoning dataset, the three-layer Transformer exhibits a sharp phase transition at approximately 800 training steps. Before this transition, the model assigns nearly uniform probabilities (around 0.2) to all possible end tokens ([END-T] and all [END-NT] tokens), albeit with a slight bias toward [END-T]. This indicates an almost random guessing behavior in this *slow learning phase*.

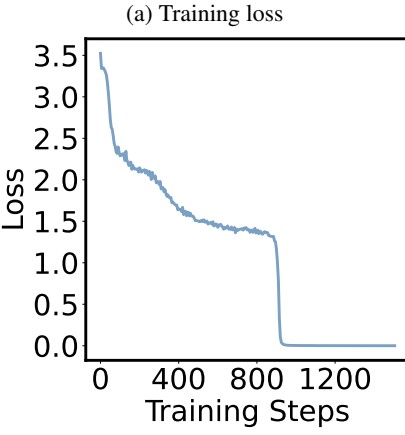
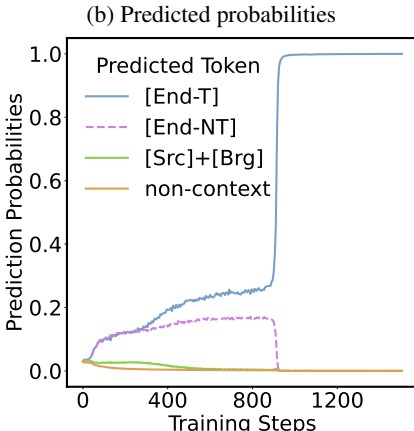

Figure 1: **The loss and the predicted probabilities.** *Left (a):* The cross entropy loss computed at the query token, with the label being the correct target end token ([END-T]) in the preceding premises. *Right (b):* The predicted probabilities for different tokens throughout training. The [END-NT] line represents probabilities averaged across [End-NT$_1$], [End-NT$_2$], [End-NT$_3$], and [Brg-NT$_4$]. Before approximately 800 steps, the [END-NT] and [END-T] lines remain close, indicating an almost random guessing behavior during this *slow learning phase*.

In contrast, single-layer and two-layer Transformers fail to consistently solve the symbolic task, even after extensive training. Additionally, we observe that removing the MLP layers from the three-layer Transformer (thus using an attention-only Transformer) does not harm the observed learning dynamics or the phase transition. Therefore, our subsequent analysis will focus on the three-layer attention-only Transformers.

## 3.2 A METHODOLOGY FOR REVERSE ENGINEERING TRANSFORMERS

In the following sections, we reverse-engineer how a three-layer attention-only Transformer learns to perform two-hop reasoning by analyzing its internal information flow during training. We start by introducing two primary methodologies for empirical analysis: examining *attention logits* and applying the *logit lens* technique.

**Attention logits.** Our first method analyzes the Transformer's attention mechanism, which controls how information flows between tokens. Specifically, we examine the *attention logits*, the raw scalar values computed immediately before the softmax operation within the attention layer. These logits quantify how strongly one token retrieves information from another. Plotting attention logits also reduces the complications induced by the softmax operation. For clarity, we refer to the values after the softmax operation as *attention weights*. We visualize attention logits across tokens and layers, highlighting entries with notably large values. When a token *attends to* another, the corresponding attention logit is high, indicating the information retrieval. This retrieval process involves copying part or all of the information from the key token into a buffer, potentially separate from the query token's own information storage.

**Logit lens.** Our second method employs the *logit lens* approach (Belrose et al., 2023), which interprets hidden states within the Transformer by projecting them directly onto the output space used for next-token prediction. In standard Transformers, the logits at the final layer are computed using a READOUT operator, which applies layer normalization followed by a linear transformation into the vocabulary space. These logits, calculated at the final token position (the query token), are then converted into probabilities through softmax.

Formally, let $h_{\text{query}}$ be the hidden state at the query token in the final layer, and ATTN-WEIGHT$(i)$ the attention weight between the $i$-th token and the query token. The probability of the next token is approximated by:

$$\text{SOFTMAX}(\text{READOUT}(h_{\text{query}})) \approx \text{SOFTMAX}\left(\sum_i \text{ATTENTIONWEIGHT}(i) \cdot \text{READOUT}(\text{VAL}(i))\right),$$

where the summation includes all preceding tokens. In the following sections, we track these vectors after the readout operator in the earlier layers to clarify how the final logits and probabilities are computed.

### 3.3 MECHANISTIC INTERPRETATION FOR THE SLOW LEARNING PHASE

Phenomenologically, during the slow learning phase (prior to the sharp phase transition), the model effectively resorts to random guessing among all possible end tokens ([END-T] and [END-NT] tokens). To explain this behavior, we closely examine attention logits across each Transformer layer.

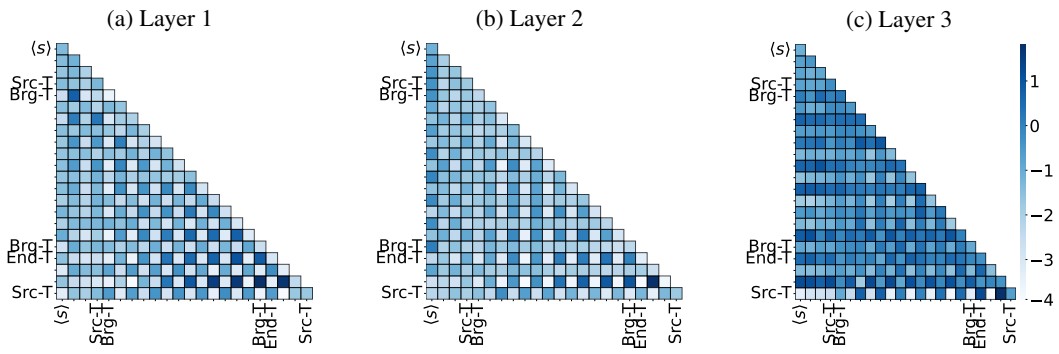

Figure 2: **Attention logits heatmaps of the three-layer Transformer at the slow learning phase (training step 800)**: *Left (a):* The first layer. Chess-board-like pattern; *Middle (b):* The second layer. The parent tokens have uniform attention on all preceding tokens. Each child token still *uniformly* attends to *all* parent tokens appearing in the preceding premises; *Right (c):* The third layer. The query token uniformly attends to *all* child tokens in the preceding premises. The query token retrieves the information from all [END] and [BRG] tokens. The logit lens could give a complete explanation for the random guessing, as shown in Figure 3.

**The first layer.** The first attention layer shows chess-board-like patterns. Each parent token attend to *all* child tokens appearing in the preceding premises, and vice versa, forming the alternating bright-dark grids on the heatmap, as shown in Figure 2a. In particular, [END] tokens (including [END-T] and [END-NT] tokens) retrieve information from [SRC] and [BRG] tokens.

**The second layer.** In the second layer, we observe a different information retrieval pattern, as presented in Figure 2b. Here, each parent token changes to attend to all preceding tokens uniformly. Each child token still *uniformly* attends to *all* parent tokens appearing in the preceding premises. Crucially, for every child token, these attention logits are *approximately equal* in magnitude. This uniform attention suggests that, at this intermediate stage, the model indiscriminately aggregates information from all prior parent tokens in the context without distinguishing between relevant and irrelevant tokens.

**The third layer.** The third layer attention logits in Figure 2c show that the query token attends to all preceding child tokens, including all [BRG] tokens in [SRC]-to-[BRG] premises and all [END] tokens in [BRG]-to-[END] premises.

**Mechanistic Explanation of random guessing.** Surprisingly, despite the final layer query token attending to all preceding child tokens, the resulting next-token probabilities strongly favor [END] tokens, resulting in the random guessing over all [END] tokens instead of all child tokens of the premises (as [BRG] tokens are also child tokens in some premises).

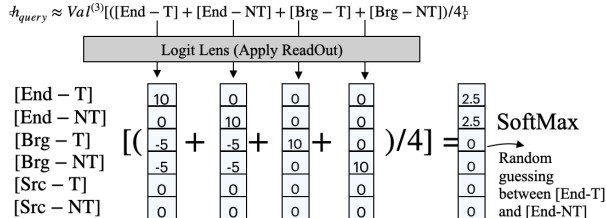

> **EXAMPLE 3: ONE DISTRACTOR**
>
> Context: <BOS> [SRC-NT]
> [BRG-NT] [SRC-T] [BRG-T]
> [BRG-NT] [END-NT] [BRG-T]
> [END-T] [SRC-T]?
>
> <BOS>: Begin-of-sequence token.
> Red: Tokens from the target chain.
> Blue: The distractor chain.
> [SRC-T]: The query token.

Figure 3: Illustration of the logit lens results of the query token at the layer 3 during the slow learning phase. The [END] tokens have positive entries at their own positions and negative entries with their preceding [BRG] tokens. These positive and negative values cancel each other out after summation.

The logit lens clearly illustrates this phenomenon. For visualization purposes, Figure 3 presents the logit lens results with a simplified two-hop reasoning task (containing only one distractor) as shown in Example 3.3. The complete numerical results are provided in Section B. In Figure 3, each [END] token has *negative entries* associated with preceding parent tokens and *positive entries* at its own position. This is probably associated with the fact that every child token attends to *all preceding parent tokens* in the second layer. Thus, the READOUT operation at the query token aggregates the value vectors equally. Formally, this is

$$\text{READOUT}(h_{\text{query}}) \approx \frac{1}{|\text{CHILD TOKENS}|} \sum_{i \in \text{CHILD TOKENS}} \text{READOUT}(\text{VAL}(i))$$

$$= \frac{1}{|\text{CHILD TOKENS}|} \left[ \sum_{i \in \text{CHILD TOKENS}} \mathbf{e}_i - \sum_{j \in \text{PARENT TOKENS}} a_j \mathbf{e}_j \right], \quad (1)$$

where $a_j > 0$ are some positive numbers, $\mathbf{e}_i$ is the vector with $i$-tn entry being one and others zero. |CHILD TOKENS| is the number of all preceding child tokens. Since some negative factors always appear at the [SRC] and [BRG] tokens' entries, the most significant entries in $\text{READOUT}(h_{\text{query}})$ are all [END] tokens. After applying softmax, all other entries will become negligible, while similar positive magnitudes for all [END] tokens yield approximately equal probabilities, resulting in random guesses over all [END] tokens.

## 3.4 MECHANISTIC INTERPRETATION FOR THE STRUCTURED LEARNING PHASE

In our experiments, after several hundred gradient steps, the three-layer Transformer successfully learns the two-hop tasks. To understand how this happens, we closely examine attention logits across each layer.

**The first layer.** The first attention layer shows a clear token-copying mechanism. As shown in Figure 4a, the only significant entries in the attention map are those connecting the paired parent token and the child token within every individual premise. Each child token copies information directly from the paired parent token. For instance, the target bridge token ([BRG-T]) retrieves information from the target source token ([SRC-T]), and the target end token ([END-T]) similarly retrieves information from the target bridge token in [BRG-T]-to-[END-T] premise.

**The second layer.** In the second layer, the only significant attention occurs primarily between the query token and the target bridge token ([BRG-T]). This strong attention indicates that the query token retrieves important information directly from the [BRG-T] token. Since the [BRG-T] token already contains information from its parent (the [SRC-T] token) due to the first-layer copying, the query token now possesses information from both [SRC-T] and [BRG-T] tokens. This targeted retrieval is

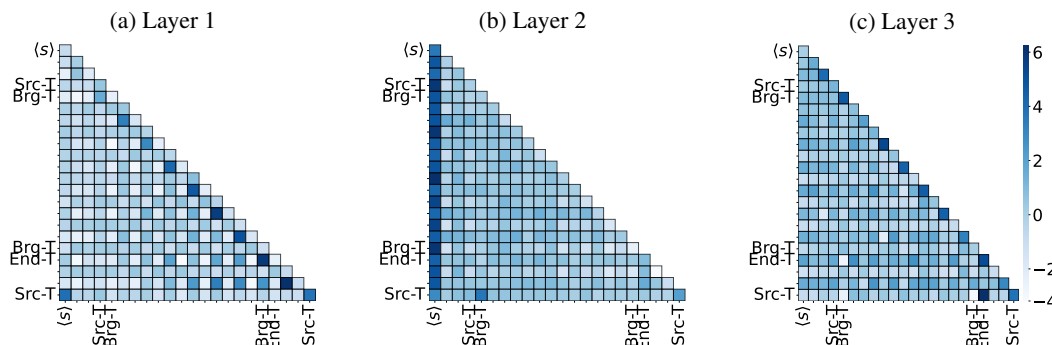

Figure 4: **Attention logits heatmaps of the three-layer Transformer at the structured learning phase (training step 10000)**: *Left (a):* The first layer. Each child token strongly attends to its parent token; *Middle (b):* The second layer. The query token strongly attends to the target bridge token ([BRG-T]); *Right (c):* The third layer. The query token strongly attends to the target end token ([END-T]). The query retrieve the identity of the target end token ([END-T]), enabling the correct next-token prediction.

possible because the query and [BRG-T] tokens share the same information (of the [SRC-T] token, since the [BRG-T] token attends to [SRC-T] from the first layer, while the query token retrieves [SRC-T] in the second layer), allowing them to effectively attend to each other.

**The Third Layer.** In the final layer, attention exclusively connects the query token and the target end token ([END-T]). This selective attention arises because both tokens share information from the target bridge token ([BRG-T]): the [END-T] token received this information directly from [BRG-T] in the first layer, while the query token obtained it indirectly through the second layer. Consequently, the final attention mechanism precisely aligns the query token with the [END-T] token, resulting in a next-token prediction that distinctly favors the [END-T] token. This structured attention ensures the model's predictions are both accurate and interpretable.

**A simplified three-layer model.** To understand the relationship between the observed mechanisms and the loss training dynamics of the three-layer Transformer. We further build a three-parameter model that essentially captures both the observed mechanisms and the loss dynamics of Transformers, building strong connections between them. We relegate the details to Appendix C.

## 4 CONCLUSIONS

In this paper, we study the underlying mechanism that transformer-based LLMs use to solve in-context two-hop reasoning tasks, especially in the presence of distracting information. We synthesize a novel dataset and find that many large pre-trained models are vulnerable to two-hop reasoning with distraction and may perform the uniform guessing mechanism, and very few steps of fine-tuning suffice to teach the model to learn a correct mechanism. By carefully analyzing the training dynamics and fully reverse-engineering a three-layer Transformer, we identified the random guessing mechanism during the early training stages and the structured learning after a sharp phase transition. Our work could bring new insights into the internal reasoning mechanisms of LLMs. Extending our work to multi-hop reasoning and more LLMs would be important future work.

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
