# HOW DO TRANSFORMERS PERFORM TWO-HOP REASONING IN CONTEXT?

## ABSTRACT

"Socrates is human. All humans are mortal. Therefore, Socrates is mortal." This form of argument illustrates a typical pattern of two-hop reasoning. Formally, two-hop reasoning refers to the process of inferring a conclusion by making two logical steps, each connecting adjacent concepts, such that the final conclusion depends on the integration of both steps. It is one of the most fundamental components of human reasoning and plays a crucial role in both formal logic and everyday decision-making. Despite recent progress in large language models (LLMs), we surprisingly find that they are vulnerable when distractors are present. We observe on a synthetic dataset that pre-trained LLMs often resort to random guessing among all plausible conclusions. However, after few steps of fine-tuning, models achieve near-perfect accuracy and exhibit strong length generalization. To understand the underlying mechanisms, we train a 3-layer Transformer from scratch on a synthetic two-hop reasoning task and reverse-engineer its internal information flow. We observe a clear progression in the attention logits throughout training. This pictures a sharp phase transition from an initial stage of random guessing to the emergence of a structured sequential query mechanism, where the model first retrieves the preceding and the bridge concepts in the early layers and then uses them to infer the final answer. Finally, we show that these dynamics can be captured by a minimal three-parameter attention-only network.

## 1 INTRODUCTION

Modern Large Language Models (LLMs) trained on language data have shown impressive abilities in solving complex reasoning tasks, including linguistic, mathematical, and programming problems (Cobbe et al., 2021; Wei et al., 2022; Achiam et al., 2023; Dubey et al., 2024; Yang et al., 2024a; Shao et al., 2024; Guo et al., 2025). Among these, multi-hop reasoning stands out as one of the most fundamental and important forms of reasoning. It refers to the process of drawing conclusions by integrating information across multiple intermediate steps or pieces of evidence. For example, consider the following chain of facts: *Marie Curie was a physicist, physicists study matter, and matter is fundamental to understanding the universe*. From these premises, we can infer that *Marie Curie contributed to understanding the universe*. This reasoning requires multiple inferential steps, each built on the previous one, making the reasoning process *inherently compositional*. Such multi-hop reasoning is critical for tasks where information must be integrated across several logical or semantic connections. Despite its simplicity and clear structure, it still lacks a clear understanding of *whether and how* pre-trained LLMs can *reliably* perform such reasoning.

In this paper, we study the mechanism behind **two-hop reasoning**, which is the simplest and most tractable form of multi-hop reasoning. We define a premise as a factual statement that connects two entities through a specific relation. In two-hop reasoning, two such premises form a chain that allows one to infer a final conclusion. We define the source entity ([SRC]) as the starting point, the bridge entity ([BRG]) as the intermediate link, and the end entity ([END]) as the target of inference. Here is a

---

[0]Our code is available at the anonymous repository https://anonymous.4open.science/r/twohopIC-2FEC/README.md.

concrete example that illustrates the above reasoning procedure:

$$\text{(Premises)} \quad \underbrace{\text{Socrates}}_{[\textsc{Src}]} \text{ is } \underbrace{\text{human}}_{[\textsc{Brg}]} . \quad \underbrace{\text{Humans}}_{[\textsc{Brg}]} \text{ are } \underbrace{\text{mortal}}_{[\textsc{End}]} .$$

$$\text{(Conclusion)} \quad \underbrace{\text{Socrates}}_{[\textsc{Src}]} \text{ is } \underbrace{\text{mortal}}_{[\textsc{End}]} .$$

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

# A  ADDITIONAL DETAILS IN LLM EXPERIMENTS

## A.1  LLM EVALUATION DETAILS

We design six two-hop reasoning templates with the aid of Deepseek-R1 (Guo et al., 2025). We choose common male and female names. To avoid interference by prior knowledge, we generate fake locations and biological categories.

---

**Templates for Two-hop Reasoning with Distractors**

**Family relations**: "[A] is the mother of [B]. [B] is the mother of [C]. Therefore, [A] is the grandmother of"
"[A] is the father of [B]. [B] is the father of [C]. Therefore, [A] is the grandfather of",
**Geographical relations**: "[A] is a city in the state of [B]. The state of [B] is part of the country [C]. Therefore, [A] is located in"
"[A] lives in [B]. People in [B] speak [C]. Therefore, [A] speaks",
**Biological relations**: "[A] is a species in the genus [B]. The genus [B] belongs to the family [C]. Therefore, [A] is classified under the family",
**Arithmetic relations**: "[A] follows the time zone of [B]. [B] is three hours ahead of [C]. Therefore, [A] is three hours ahead of",

---

**Entity names**

**Locations**: "Zorvath", "Tyseria", "Kryo", "Vynora", "Quellion", "Dras", "Luminax", "Vesperon", "Noctari", "Xyphodon", "Glacidae", "Ophirion", "Eryndor", "Solmyra", "Umbrithis", "Balthorien", "Ytheris", "Fendrel", "Havroth", "Marendor"
**Biological taxonomy**: "Fluxilus", "Varnex", "Dranthidae", "Zynthor", "Gryvus", "Myralin", "Thalorium", "Zephyra", "Aerinth", "Xyphodon", "Kryostis", "Glacidae", "Borithis", "Chrysalix", "Noctilura", "Phorvian", "Seraphid", "Uthrelin", "Eldrinth", "Yvorith"
**Languages**: "English", "Spanish", "Mandarin", "Hindi", "Arabic", "French", "German", "Japanese", "Portuguese", "Russian", "Korean", "Italian", "Turkish", "Dutch", "Swedish", "Polish", "Hebrew", "Greek", "Bengali", "Thai"
**Names**: "Ben", "Jack", "Luke", "Mark", "Paul", "John", "Tom", "Sam", "Joe", "Max", "Amy", "Emma", "Anna", "Grace", "Kate", "Lucy", "Sarah", "Alice", "Alex", "Ruby"

---

## A.2  LLM FINETUNING DETAILS

We train four models on two-hop reasoning with **one** distractor using LoRA Hu et al. (2022). We list the training configuration details in Table 2. The experiments were completed within one hour utilizing four NVIDIA A100 GPUs.

## A.3  EVALUATION RESULTS FOR MORE MODELS

We evaluate more LLMs on the two-hop reasoning with distractors before or after the finetuning. Tables 3, 4, and 5 present the average probabilities on the answer and distractors.

# B  ADDITIONAL DETAILS IN THREE-LAYER TRANSFORMER EXPERIMENTS

## B.1  ADDITIONAL TRAINING DETAILS

We list the training details in Table 6.

Table 2: LLM Fine-tuning with LoRA Configuration

| Setting | Value |
|---|---|
| **Model Configuration** | |
| Model Names | Qwen2.5-7B, OLMo-7B, Llama2-7B, Llama3.1-8B |
| Precision | fp16 |
| **LoRA Configuration** | |
| LoRA Dimension (r) | 16 |
| LoRA Alpha Parameter | 32 |
| LoRA Dropout | 0.05 |
| Bias | None |
| Target Modules | ["q_proj", "k_proj", "v_proj", "o_proj"] |
| **Data Processing** | |
| Distractor Numbers | 2 |
| Maximum Sequence Length | 128 |
| **Optimization** | |
| Optimizer | AdamW |
| Learning rate | $2 * 10^{-4}$ |
| Learning rate scheduler | cosine |
| Warmup ratio | 0.03 |
| Weight decay | 0.01 |
| Epochs | 3 |
| Batch size | 16 |

Table 3: Performance comparison of Qwen2.5-7b base and fine-tuned models on two-hop reasoning tasks with varying numbers of distractors. $K = 1$ indicates the scenario without distractors. The [END-T] and [END-NT] rows report the average probabilities assigned to the first token of the target end entity and non-target end entities, respectively. For cases with multiple distractors, the reported probabilities for non-target entities are averaged first over all [END-NT] within each context, then across all contexts. The values in the parentheses are the standard errors.

| Models | Next Tokens | The Number of Reasoning Chains in the Context ($K$) | | | | |
|---|---|---|---|---|---|---|
| | | 1 | 2 | 3 | 4 | 5 |
| **Qwen2.5-7B** | [END-T] | 0.80 (0.01) | 0.29 (0.01) | 0.26 (0.01) | 0.26 (0.01) | 0.24 (0.01) |
| | [END-NT] | NA (NA) | 0.27 (0.01) | 0.10 (0.00) | 0.06 (0.00) | 0.05 (0.00) |
| **Fine-tuned on $K = 2$** | [END-T] | 1.00 (0.00) | 1.00 (0.00) | 1.00 (0.00) | 1.00 (0.00) | 0.99 (0.00) |
| | [END-NT] | NA (NA) | 0.00 (0.00) | 0.01 (0.00) | 0.00 (0.00) | 0.00 (0.00) |

## B.2 LOGIT LENS RESULTS

We present the details for logit lens results. Suppose that given any input sequence, the value state of token $t$ at layer 3 is VALUE$^{(3)}$. Suppose the projection matrix in the attention layer 3 is $\mathbf{O}^{(3)}$, the layer norm before the READOUT matrix is LAYERNORM. For any token $t \in \{$ [Brg$_1$], ..., [Brg$_5$], [END][1], ..., [END][5] $\}$, using the logit lens, we compute

$$\text{OUTPUT-LOGITS}(t) = \text{READOUT}(\text{LAYERNORM}(\mathbf{O}^{(3)}\text{VALUE}^{(3)}(t))).$$

We focus on the entries on the OUTPUT-LOGITS$(t)$ corresponding to tokens in $t \in \{$ [Brg$_1$], ..., [Brg$_5$], [END][1], ..., [END][5] $\}$. Figure 5 shows their logit lens results computed at the model trained with 800 steps. All tokens have positive logits for themselves. [END] tokens have negative logits on [BRG] tokens. [BRG] tokens have approximately zero logits on [END] tokens.

Table 4: Performance comparison of Llama3.1-8B base and fine-tuned models on two-hop reasoning tasks with varying numbers of distractors. $K = 1$ indicates the scenario without distractors. The [END-T] and [END-NT] rows report the average probabilities assigned to the first token of the target end entity and non-target end entities, respectively. For cases with multiple distractors, the reported probabilities for non-target entities are averaged first over all [END-NT] within each context, then across all contexts. The values in the parentheses are the standard errors.

| Models | Next Tokens | The Number of Reasoning Chains in the Context $(K)$ | | | | |
|---|---|---|---|---|---|---|
| | | 1 | 2 | 3 | 4 | 5 |
| **Llama3-8b** | [END-T] | 0.90 (0.00) | 0.40 (0.01) | 0.32 (0.01) | 0.29 (0.01) | 0.27 (0.01) |
| | [END-NT] | NA (NA) | 0.37 (0.01) | 0.19 (0.00) | 0.13 (0.00) | 0.10 (0.00) |
| **Fine-tuned on $K = 2$** | [END-T] | 1.00 (0.00) | 1.00 (0.00) | 1.00 (0.00) | 1.00 (0.00) | 1.00 (0.00) |
| | [END-NT] | NA (NA) | 0.00 (0.00) | 0.00 (0.00) | 0.00 (0.00) | 0.00 (0.00) |

Table 5: Performance comparison of Llama2-7b base and fine-tuned models on two-hop reasoning tasks with varying numbers of distractors. $K = 1$ indicates the scenario without distractors. The [END-T] and [END-NT] rows report the average probabilities assigned to the first token of the target end entity and non-target end entities, respectively. For cases with multiple distractors, the reported probabilities for non-target entities are averaged first over all [END-NT] within each context, then across all contexts. The values in the parentheses are the standard errors.

| Models | Next Tokens | The Number of Reasoning Chains in the Context $(K)$ | | | | |
|---|---|---|---|---|---|---|
| | | 1 | 2 | 3 | 4 | 5 |
| **Llama2-7b** | [END-T] | 0.82 (0.01) | 0.41 (0.01) | 0.31 (0.01) | 0.25 (0.01) | 0.20 (0.01) |
| | [END-NT] | NA (NA) | 0.34 (0.01) | 0.20 (0.01) | 0.14 (0.00) | 0.11 (0.00) |
| **Fine-tuned on $K = 2$** | [END-T] | 1.00 (0.00) | 1.00 (0.00) | 0.92 (0.01) | 0.89 (0.01) | 0.86 (0.01) |
| | [END-NT] | NA (NA) | 0.00 (0.00) | 0.04 (0.01) | 0.03 (0.01) | 0.03 (0.00) |

## C  THE THREE-PARAMETER MODEL

**"Causal" hypotheses based on observations.**    In Section 3, we observe two stages along the training dynamics of the three-layer transformer. In this section, we aim to build a "causal" relationship between the observed mechanisms and the training dynamics. The two "causal" hypotheses are:

**Hypothesis C.1.** *The formation of the random guessing mechanism causes the slow learning phase (0-800 steps).*

**Hypothesis C.2.** *The formation of the sequential query mechanism causes the abrupt phase transition (800-10000 steps).*

We need to implement "causal interventions" to validate the hypotheses. Following the approach of Reddy (2023), we propose studying a *three-parameter dynamical system*, which simulates only the dynamics of the sequential query mechanism, removing the random guessing mechanism.

**Comparing the training dynamics of the three-parameter model with the training dynamics of transformers to validate the causal hypotheses**    Since the random guessing mechanism is removed and the sequential query mechanism is kept, we anticipate that

1. Hypothesis C.1 holds if the three-parameter model loses the slow learning phase in the training dynamics.

2. Hypothesis C.2 holds if the three-parameter model preserves the abrupt phase transition in the training dynamics.

Table 6: Transformer Training Configuration

| Setting | Value/Default |
|---|---|
| **Model Architecture** | |
| Embedding Type | Positional embeddings |
| Normalization | Pre-layer normalization |
| MLP Activation | ReLU in mlp |
| **Optimization Parameters** | |
| Optimizer | Adam |
| Learning Rate | 0.0003 (fixed) |
| $\beta_1$ | 0.9 |
| $\beta_2$ | 0.99 |
| $\epsilon$ | $10^{-8}$ |
| Weight Decay | 0.01 |
| **SGD Configuration** | |
| Learning Rate (3-parameter model) | 0.1 |
| **Training Details** | |
| Batch Size ($B$) | 512 |
| Sequence Length ($N$) | 23 |
| Training Steps | 10,000 |
| Seeds | Results are consistent across different random seeds |

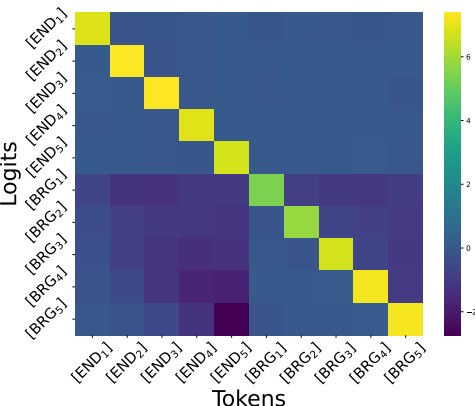

Figure 5: **Logit lens of the value states at the layer** 3: We use the model trained on step 800. The results are averaged over 256 sequences. The $y$-axis represents the entries on the logit lens output. The $x$-axis represents tokens. The bright color indicates larger value, and blue color indicates negative values. The bright diagonal line shows that all tokens have value states that strongly support predicting themselves. The left bottom blue part indicates that [END] tokens have negative values for [BRG] tokens.

**Approximate SoftMax operator and notations for content and buffer spaces.** For convenience, we define the approximate SoftMax operator as:

$$\widetilde{\mathcal{S}}(u, M) = \frac{\exp(u)}{\exp(u) + M}.$$

Intuitively, the approximate SoftMax gives the probability of an item with logit $u$ where the remaining $M$ logits are all zero. Given a residual state $u$, we use $\text{CONT}(u)$, $\text{BUF}_1(u)$, $\text{BUF}_2(u)$ to denote the original content (i.e., the token embedding and the positional embedding) of token $u$, the buffer space of $u$ in the first layer, and the buffer space of $u$ in the second layer, respectively. Since the tokens are not semantically related, we assume that $\langle \text{CONT}(a), \text{CONT}(b) \rangle = \mathbb{1}\{a = b\}$ for any two tokens $a$ and $b$, which means that $\{\text{CONT}(\cdot)\}$ is an orthonormal basis.

**The meaning of three parameters.** The sequential query mechanism consists of a copy layer in the first layer, query to $\text{BRG}$ in the second layer, and query to [END] in the third layer. We use parameters $\alpha$, $\beta$, and $\gamma$ to represent the progressive measure of their functionalities.

$$\text{BUF}_1([\text{BRG-T}]) = w_1 \text{CONT}([\text{SRC-T}]), \tag{2}$$

$$\text{BUF}_1([\text{END-T}]) = w_1 \text{CONT}([\text{BRG-T}]), \tag{3}$$

$$\text{BUF}_2(\text{query}) = w_2 \text{CONT}([\text{BRG-T}]), \tag{4}$$

$$\text{Output} = w_3 \text{CONT}([\text{END-T}]), \tag{5}$$

$$\text{Loss} = -\log[\widetilde{\mathcal{S}}(\xi w_3, V)]. \tag{6}$$

where

$$w_1 = \widetilde{\mathcal{S}}(\alpha, N),$$

$$w_2 = \widetilde{\mathcal{S}}(\beta \langle \text{CONT}(\text{query}), \text{BUF}_1([\text{BRG-T}]) \rangle, 2N),$$

$$w_3 = \widetilde{\mathcal{S}}(\gamma \langle \text{BUF}_2(\text{query}), \text{BUF}_1([\text{END-T}]) \rangle, 2N).$$

Note that when we set $\alpha \to \infty$, $\beta \to \infty$, and $\gamma \to \infty$, $\text{Loss} \to 0$, corresponding to the three-layer transformer trained after 10000 steps. When we set $\alpha = \beta = \gamma = 0$, the loss is close to a uniform guess in the vocabulary, corresponding to an untrained three-layer transformer.

**The derivation of Equations** (2) **and** (3). We present how we simplify a full transformer block to get Equations (2) and (3). As illustrated in Section 3, the first attention block relies on the positional information to copy parent tokens to the buffer spaces of child tokens. The attention logits are given by

$$\text{Attn-Logit}(\text{CHILD} \to \text{PARENT})$$
$$= \text{Pos}_i^\top Q^{(1)\top} K^{(1)} \text{Pos}_{i-1},$$

where $Q^{(1)}$, $K^{(1)}$ are weight matrices in the first layer. We assume that $\text{Pos}_i^\top Q^{(1)\top} K^{(1)} \text{Pos}_{i-1} = \alpha$ for any $i$. Since we reshuffle the positions for $[\text{Brg}_1]$ and [END] for each sequence, following Reddy (2023), we approximate the attention weights to parent tokens by $\widetilde{\mathcal{S}}(\alpha, N)$, where $N$ comes from taking the average from $2N$ positions. This gives Equations (2) and (3).

**The derivation of Equation** (4). Similarly, the $\text{BUF}_2(\text{query})$ is proportional to the attention from the query token to [BRG-T] in the second layer. The query token uses its $\text{CONT}(\text{query})$ to fit the $\text{BUF}_1([\text{BRG-T}])$, copying $\text{CONT}([\text{BRG-T}])$ to the residual stream. Therefore,

$$\text{Attn-Logit}(\text{query} \to [\text{BRG-T}])$$
$$= \text{CONT}(\text{query})^\top Q^{(2)\top} K^{(2)} \text{BUF}_1([\text{BRG-T}])$$
$$= \beta \cdot \langle \text{BUF}_1([\text{BRG-T}]), \text{CONT}(\text{query}) \rangle,$$

where the last line could be viewed as a re-parametrization of $Q^{(2)\top} K^{(2)}$, with $\beta \propto \|Q^{(2)\top} K^{(2)}\|_2$. Moreover, we fix the attention logits from query token to all other tokens to be zero, removing mechanisms other than the sequential query. The attention weight from the query token to the [BRG-T] becomes $\widetilde{\mathcal{S}}(\text{Attn-Logit}(\text{query} \to [\text{BRG-T}]), 2N)$. This gives Equation (4).

**The derivation of Equation** (5) **and** (6). The query token increasingly concentrates on the [END-T] token along the training dynamics. With the same manner of Equation (4), we set that

$$\texttt{Attn-Logit}(\text{query} \to [\text{END-T}])$$
$$= \text{BUF}_2(\text{query})Q^{(3)\top}K^{(3)}\text{BUF}_1([\text{END-T}])$$
$$= \gamma \cdot \langle \text{BUF}_2(\text{query}), \text{BUF}_1([\text{END-T}]) \rangle.$$

We focus on $\texttt{Attn-Logit}(\text{query} \to [\text{END-T}])$ and set all other $\texttt{Attn-Logit}$ to be zero. The attention weight from the query to the [END-T] becomes $\widetilde{S}(\texttt{Attn-Logit}(\text{query} \to [\text{END-T}]), 2N)$. We first consider the output $\texttt{Logit}$ on the query token. Through the logit lens, as illustrated in Figure 5, the value states of [END-T] tokens have large logits on themselves. Therefore, we assume that $\texttt{ReadOut}[\text{VAL}([\text{END-T}])] = \xi \cdot e_{[\text{END-T}]} \in \mathbb{R}^V$, with $\xi > 0$ and $e_{[\text{END-T}]}$ being a one-hot vector in $\mathbb{R}^V$ that is non-zero on the index of [END-T]. In our simulation, we fix $\xi = 30$. The loss can therefore be approximated through Equation (6).

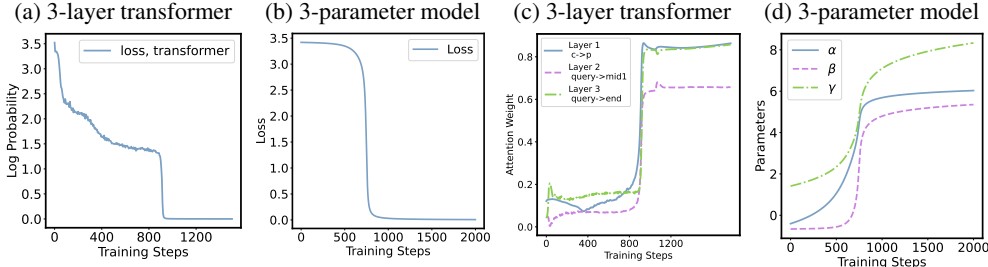

(a) 3-layer transformer  (b) 3-parameter model  (c) 3-layer transformer  (d) 3-parameter model

Figure 6: **The comparison between 3-layer transformer and 3-parameter model validates our causal hypotheses.** *Top left (a)*: The loss dynamics of the 3-layer transformer shows a slow learning phase (0-400 steps) followed by an abrupt phase transition (around 800 steps). *Top right (b)*: The 3-parameter model, which only simulates the sequential query mechanism, skips the slow learning phase but preserves the abrupt phase transition, validating both hypotheses. *Bottom left (c)*: The dynamics of important attention weights for the sequential query mechanism. *Bottom right (d)*: The parameter dynamics of the 3-parameter model shows synchronized phase transitions in all three parameters ($\alpha, \beta, \gamma$), indicating the formation of the sequential query mechanism.

**Simulations on three-parameter model validate Hypotheses C.1 and C.2.** We optimize the loss function in Equation (6) by gradient descent with learning rate $0.1$. Figure 6 presents the training dynamics of the 3-layer transoformer and the 3-parameter model. Since the model does not incorporate the random guessing mechanism, the loss remains unchanged during the first $1000$ steps, validating Hypothesis C.1. Both the parameters and the loss function go through a sudden phase transition around step $1000$, suggesting that the emergence of the sequential query mechanism is the driving force behind the abrupt drop in loss. This validates Hypothesis C.2.