# OpenReview forum: "How Do Transformers Perform Two-Hop Reasoning in Context?"
_ICLR.cc/2026/Conference — Submitted to ICLR 2026_

### Official Review · Reviewer_BMFe · 2025-10-17

**Soundness:** 2
**Presentation:** 3
**Contribution:** 2
**Rating:** 4
**Confidence:** 3

**Summary:**

This paper studies how Transformers and large language models (LLMs) perform two-hop (or transitive) reasoning in context. The first part evaluates pretrained and fine-tuned LLMs on a synthetic “two-hop reasoning with distractors” dataset. The authors observe that pretrained models behave like random guessers when distractors are present, but quickly reach near-perfect accuracy after minimal fine-tuning. The second part examines a small three-layer Transformer trained from scratch on a symbolic version of the same task, performing mechanistic interpretability analyses of attention patterns and information flow. The authors identify a phase transition between a “random guessing” and a “structured sequential query” phase, capturing this transition with a simple three-parameter analytical model.

**Strengths:**

- The paper is clearly written and well structured.
- The mechanistic analysis of a small Transformer is detailed and replicable.
- The results on phase transitions and the identification of structured attention mechanisms are interesting and consistent with trends observed in mechanistic interpretability research.
- The connection between the behavioral LLM findings and the toy-model analysis is conceptually appealing.

**Weaknesses:**

1. Limited originality of the first part.
The analysis of pretrained models’ failure on synthetic two-hop reasoning tasks seems to replicate findings that are already known. Previous work—such as Guzmán et al. (Findings of NAACL 2022) and the recent preprint [arXiv:2505.14313]—already explores logical and transitivity reasoning in LLMs with more formal frameworks and broader coverage. Compared to these, the current contribution feels incremental. The framing as “two-hop reasoning” does not clearly extend the conceptual understanding of reasoning failures beyond those studies. The “two-hop” framing would benefit from clearer alignment with classical notions of transitivity or syllogistic reasoning, to situate the work within the broader reasoning literature.

2. Questionable assumptions in the dataset design.
The synthetic dataset implicitly assumes that the model can recognize that the relations involved are transitive. However, this assumption is not tested or controlled for. If the model does not in fact represent these relations as transitive, then its apparent failure may not reflect limits in reasoning but rather a misunderstanding of the relational semantics themselves. The conclusions about reasoning ability could thus be overstated.

3. Potential confounds from heuristic or world knowledge effects.
The paper assumes that models rely purely on formal reasoning, but no controls are introduced to rule out heuristic shortcuts or content-based biases. Since the templates use interpretable relations (e.g., family or geography), models could exploit distributional or semantic cues rather than performing genuine logical inference. This undermines the claim that the study isolates reasoning mechanisms.

4. Unclear contribution of the mechanistic findings for improving models.
The second part convincingly documents how a small Transformer learns to perform two-hop reasoning through sequential attention patterns. However, it remains unclear what actionable insight this offers for improving LLM reasoning abilities. Does understanding this phase transition suggest concrete architectural or training modifications? The discussion does not address how such mechanistic findings could scale up or inform interventions in large models.

**Questions:**

How does your analysis of two-hop reasoning extend beyond prior studies on transitivity and logical reasoning (e.g., Guzmán et al., 2022; arXiv:2505.14313)?

Have you verified that the models actually interpret the relations in your dataset as transitive, rather than as arbitrary symbolic associations?

Could the observed behavior reflect heuristic or world-knowledge shortcuts rather than genuine logical reasoning? How did you control for such effects?

What practical implications do your mechanistic findings have for improving reasoning in larger language models or guiding model design?

How does your notion of “two-hop reasoning” relate to classical transitivity or syllogistic reasoning, and why did you choose this framing?

---

### Official Review · Reviewer_Hs31 · 2025-10-17

**Soundness:** 1
**Presentation:** 3
**Contribution:** 1
**Rating:** 2
**Confidence:** 4

**Summary:**

The paper studies two-hop reasoning with distractors. Empirically, LLMs (e.g., OLMo-7B) guess uniformly over candidate ends when multiple two-hop chains co-occur; brief fine-tuning yields near-perfect accuracy and length generalization.
To explain this, the authors train and reverse-engineer a 3-layer attention-only Transformer on a symbolic task, observing a phase transition from random guessing to a sequential query mechanism: layer-1 copies parent→child, layer-2 retrieves the bridge tied to the source, layer-3 aligns the query to the correct end. They also sketch a minimal 3-parameter analytical model capturing the loss/attention dynamics.

**Strengths:**

- The sequential-query circuit is intuitively appealing and matches observed attention maps, which is not surprising.

- Figures and explanations are pedagogically strong, easy to follow, even for OOD readers.

**Weaknesses:**

- The dataset itself introduces a shortcut bias that undermines the very reasoning claims. In other words, a synthetic symbolic dataset is too clean and shortcut-prone. Such design encourages shortcut learning (memorizing surface token positions or n-gram spans) rather than genuine compositional reasoning.

- Strong mechanistic write-up and neat minimal model, but conceptual novelty is limited relative to existing multi-hop analyses and depth theory; external validity beyond symbolics is unclear. Strengthening comparisons, adding tuned-lens checks, and demonstrating transfer to natural prompts would move this toward acceptance.

- Toy 3-layer model offers limited insight into real decoder-only LLMs. The analyzed model is trained from scratch on synthetic data and lacks positional embedding and other complex architecture designs, as well as no noisy pretraining data.
It is therefore unclear whether the observed “sequential query” reflects processes in real decoder-only models like Llama or Gema, which are trained on much more noise and complex data with many more layers.

- Empirical depth claim not grounded in formal theory. Also, the statement that “three layers are minimal for two-hop reasoning” is only observed empirically, or more like accommodates explaining the findings.

**Questions:**

none

---

### Official Review · Reviewer_i2mk · 2025-10-27

**Soundness:** 2
**Presentation:** 2
**Contribution:** 2
**Rating:** 2
**Confidence:** 4

**Summary:**

This paper analyzes how a Transformer-like toy model solves a two-hop syllogistic reasoning task.
For example, when given an input like "John is the father of Paul. Luke is the father of Tom. Paul is the father of Ben. John is the grandfather of", the model should answer "Ben", since John is the father of Paul and Paul is the father of Ben.
The paper first evaluates the performance of a 7-billion parameter LLM on this task, finding that finetuning consistently increases the output probability of the correct answer.
The main part of the paper is the analysis of a three-layer self-attention-only Transformer toy model, which the authors show is sufficient perform this task.
The analysis mainly consists of interpretation of attention maps of the three layers, supplemented with logit lens results.
While the paper claims to "reverse-engineer how a three-layer attention-only Transformer learns to perform two-hop reasoning" (line 310), the presented analysis, while providing some insights, unfortunately falls short of answering its titular question (see weaknesses).

**Strengths:**

The proposed setup using a simple model in a controlled setting could allow future work to achieve the claimed goal of "reverse engineering" how transformers perform two-hop reasoning.

**Weaknesses:**

My main concern is that the presented analysis is not deep enough to warrant the claim of having "reverse-engineered", i.e., having achieved a thorough understanding of, how Transformers perform two-hop reasoning.

More specifically, the analysis predominantly relies on attention maps, which show the degree to which information flows from source positions to target positions at each of the three layers. While this kind of analysis can be informative, it paints only part of the picture, since it does not tell us what kind of information flows and how it influences downstream components and ultimately model output. The field of (mechanistic) interpretability has developed many methods for answering such questions, e.g., circuit analysis (e.g., https://arxiv.org/abs/2305.00586) , SAEs (https://arxiv.org/abs/2309.08600), sparse feature circuits (https://arxiv.org/abs/2403.19647), causal interventions, etc.

As it is, the analysis feels more like a collection of assorted observations than a full account of how the examined toy model performs two-hop reasoning.

**Questions:**

Minor comments:

1. About line 116: "a Transformer with log k layers is both necessary and sufficient to perform k-hop reasoning in context (Sanford et al., 2024b)."
Since log k < 3 for k = 2, this would be a contradiction to "the minimal architecture required is a three-layer Transformer." (line 115). According to Sanford et al., 2024b (Theorem 4.2), the number of layers is L = ⌊log_2 k⌋ + 2

2. About Section 2.2:
In Example 1, the base model gives the highest probability to the correct answer, so unless I am misunderstanding something, the base model performs the task correctly, even in the presence of distractors.
What the Example 1 and the evaluation in Table 1 do not show is if and how much task accuracy decreases as the number of distractors increases, both for the base model and the finetuned model.
I think this would be important, since the current evaluation, which analyzes only probabilities, does not support the claim that "fine-tuning significantly enhances LLM robustness and generalization capabilities." (line 219). The current evaluation only shows that the output probability distributions become sharper, i.e., more probability mass is assigned to the correct output token. Yes, a large margin between the correct and wrong token probabilities might be desirable, but a model could also generalize perfectly if it consistently assigns 0.01 higher probability to the correct token, as in Example 1.

3. About line 324 and following: This subsection discusses the "logit lens approach", but cites Belrose et al. 2023, namely the work that intoduced the *Tuned* Logit Lens, which is a variant of Logit Lens. The canonical citation for the standard (non-tuned) logit lens is nostalgebraist, 2020. If the submission used the tuned logit lens, then this should be made clearer, and if not, I would suggest to change the citation in order to prevent misunderstandings.

4. The formula on line 334 appears out of the blue, since the start of the paragraph ("Formally, ...") is not well connected to the previous two paragraphs. In the "Attention logits" paragraph (lines 315-323), the point is made that, due to the softmax, attention logits are easier to handle than attention weights, but then the formula uses weights, only for Figure 2 to show attention logits. I can of course make an educated guess how Figure 2, the formula on line 334, and the two paragraphs from lines 315-329 are connected, but this should be made much clearer.

---

### Official Review · Reviewer_GopB · 2025-11-04

**Soundness:** 2
**Presentation:** 3
**Contribution:** 2
**Rating:** 2
**Confidence:** 3

**Summary:**

This paper investigates how transformers perform two-hop reasoning, a form of reasoning where information from two different sources need to be combined to answer a query. For this, the paper proposes to use a simplification of the syllogistic logic as the formal reasoning framework and created synthetic datasets for this. The dataset is used to investigate a transformer architecture behaviour when trained upon it. The paper reports an interesting phenomenon where the model, over the course of the training, sharply goes from random guessing to perfect accuracy.

**Strengths:**

The paper is well written and the identified phenomenon is interesting.

**Weaknesses:**

The evidence provided in the analysis section does not seem conclusive, but rather suggestive. For conclusive analyses I would like to see some causal relationships, counterfactual experiments, etc.
I don't think the behaviour observed is unique to the problem space but rather common to all synthetic datasets with a clear decision rule. It would be good to see whether the behaviour differs for other types of datasets.

I cannot comment on the novelty of the interpretability methodology or the findings (though as practitioner, they seem rather obvious to me). However, I can definitely comment that the dataset creation methodology and the investigation of its underlying reasoning processes required to solve the problems is not novel - in fact it's (a simplification of) the very first syllogistic logic and its verbalisation, the $\symfrak{E}_0$ fragment, investigated by [Pratt-Hartmann, (2008)](https://link.springer.com/article/10.1023/b:jlli.0000024735.97006.5a). The ability of transformers to reason in it (and many other more complex fragments) have been investigated as well [see here](https://arxiv.org/pdf/2211.05417), [and here](https://ojs.aaai.org/index.php/AAAI/article/view/6397).

Finally, there is no appendix which the paper suggests has additional experiments.

**Questions:**

Could the authors please do comparative analyses to other synthetic formal reasoning datasets to see if the observed behaviour is unique to two-hop reasoning and provide a causal explanation of the observed behaviour.

---

### Meta-Review · Area_Chair_xJ7P · 2026-01-07

**Summary:**

This paper studies two-hop reasoning in a controlled synthetic syllogistic setting and reports a sharp training-phase transition from near-random guessing to perfect accuracy, supplemented by attention-map and logit analysis of a three-layer Transformer model. The phenomenon itself is plausible and the paper is clearly written, but multiple reviewers find the central mechanistic claim ("reverse-engineered" two-hop reasoning) overstated given the largely descriptive analysis and lack of causal or counterfactual evidence. Concerns also remain about novelty, external validity (synthetic shortcut-prone data and a toy architecture trained from scratch), and unclear implications for real decoder-only LLMs.

The authors did not engage in discussion.

**Reviewer Concerns:**

None of the concerns were addressed.

**Reviewer Scores:**

N/A

---

### Decision · Program_Chairs · 2026-01-26

Reject